# Two Mediterranean Church Mothers: Their Presence and Importance in Patristic Philosophy

Ekin Kaynak Iltar [1,*], Rabia Akçoru [2] , Emine Atmaca [3], Nihal Kubilay Pınar [4] and Ali Bilge Öztürk [1]

1 Faculty of Letters, Philosophy, Akdeniz University, Antalya 07058, Türkiye; alibilgeozturk@akdeniz.edu.tr
2 Mediterranean Civilizations Research Institute, Mediterranean Medieval Studies, Akdeniz University, Antalya 07058, Türkiye; rabiakcoru@gmail.com
3 Faculty of Letters, Turkish Language and Literature, Akdeniz University, Antalya 07058, Türkiye; eatmaca@akdeniz.edu.tr
4 Faculty of Letters, German Studies, Akdeniz University, Antalya 07058, Türkiye; nihalpinar@akdeniz.edu.tr
* Correspondence: ekinkaynak@akdeniz.edu.tr

**Abstract:** There are many debates concerning the existence and possibility of Christian philosophy, as well as Jewish and Islamic philosophy. Starting from the debates in France in the 1930s, some philosophers, such as Bréhier, have argued that Christian philosophy, especially in the patristic era, did not operate through reason, as philosophy requires, for religion is a system full of dogmas, interpretations based on strict approaches and alleged facts. However, patristic philosophers, especially some pro-philosophy apologetic writers, suggested that pagan philosophy, which is assumed to operate through reason, took its doctrines from Christianity and distorted them. Thus, for them, Christianity was the true philosophy. If one takes the patristic philosophers' view and considers Christianity a philosophy through their writings, one should acknowledge that, at the basis of their views, lies the meaning of $\varphi\iota\lambda o\sigma o\varphi\acute{\iota}\alpha$, that is, the love of wisdom. Our consideration of early Christian philosophy as a way of life is best understood through the lens of the Church mothers since they are portrayed as the best examples among Christians. In this study, we aim to discuss patristic philosophy as the basic form of philosophizing and then go on to specify prominent female Christian philosophers from the Mediterranean Region, the crucible for many long-lasting civilizations, and finally examine the ways they were philosophizing. Our method of research is mainly based on document and content analysis since it is qualitative research in the philosophical literature. Based on the findings of this research, we demonstrate the core endeavor of philosophy as the love of wisdom through the ideas of two of the prominent women in early Christian philosophy.

**Keywords:** early Christian philosophy; women Christian philosophers; Middle Ages; Church mothers





## 1. Introduction

What is philosophy? This question may be answered with an etymological argument suggesting that the word "philosophy", from the Ancient Greek *philosophia*, consists of a combination formed with $\varphi\iota\lambda\varepsilon\widehat{\iota}\nu$ (philein), meaning "to love", and $\sigma o\varphi\acute{\iota}\alpha$ (sophia), meaning "wisdom", in Ancient Greek and is translated as "love of wisdom". However, this meaning brings forth so many questions pointing out the need to set the boundaries of "that" wisdom or simply define it. Therefore, many definitions of philosophy have been and are still being brought forth. Yet, philosophy was a guide to life for Christians in the patristic era; it was a guide to the material world exhibiting how people should live and how they could attain happiness. From this point of view, many Christian philosophers ensured that philosophy and Christianity were reconciled. They suggested that philosophy was a tool for justifying Christianity by expressing that the activity of philosophical thinking was useful even though it belonged to the Graeco-Roman pagan world. Moreover, some early Christian philosophers, such as Justin the Martyr, used the word "philosopher" for the word "Christian". In his first apology, he defines different Greek philosophical schools

with the title "philosopher", and comparing this to Christians, he assumes that Christians use the title "Christian" for all different interpretations of Christianity (I.7.3). He repeatedly compares Christians to philosophers (I.26.6, I.46.3; II.10.1–10.2, II.10.4–10.8). However, the same logic was used by some Christians to assert that Greek philosophers had stolen the doctrines of barbarians, and so their philosophy had no relationship with Christianity. That is because, while philosophy was in search of a kind of happiness that could only be obtained in the material world, happiness for Christians was related to the other world. On the contrary, they existed in the material world for the sake of the other world where they would reach the salvation promised at the end of the material world. Therefore, philosophy related to the material world would not provide the salvation necessary for the other world. For example, Tertullian, in his *Apologeticus*, compares Christians with Greek philosophers but reaches different results. He accuses Greek philosophers of stealing the truth from Moses and his followers; without hesitation, he makes a connection between Socrates and demons through the notion of the *daemon* and claims that he listens to his demon instead of the state's gods (XLVI.5–6). Afterwards, he enumerates the differences between Christians and Greek philosophers, defining Christians as disciples of heaven, those who have an interest in life and those who deal with facts, while on the other hand defining Greek philosophers as disciples of the Greeks, those whose real interest lies with fame and those who deal with words. Moreover, he positions Greek philosophers as thieves and as people distorting the truth and using it for their own benefit (XLVI.18).

These arguments were put forward during the 2nd century and later, when Christianity was still in its formation phase. This is the period in which Christians were frequently accused and martyred by persecutors, constantly forced to defend their religion, while its followers were at the very stage of disaggregating themselves from the pagan world. The Roman Empire, which initially saw Christianity as a Jewish sect and defined it mostly as a superstition, forbade conversions to Christianity,[1] as the number of those becoming Christian increased at the beginning, so this caused Christians to have their meetings at night in their members' private homes so that they could carry out their religious rituals in the house churches[2] where they gathered secretly at night and because there were no formal Church buildings. It is probable that Christians hiding, gathering in houses at night as if their religion was a mystic cult and carrying out rituals suspected by pagans because they had no idea what they were doing laid the foundation for the charges concerning them. The most learned Christians composed various works to respond to the charges brought against them as to whether they were legal or not. These authors received higher education, including in rhetoric, when they were pagans and later converted to Christianity. Consequently, their works appear as powerful examples of rhetoric. In these texts—apologies (apologia)—authors often described themselves as philosophers and their religion as philosophy. Although religion and philosophy seem like two separate fields from a modern perspective, this separation had not yet taken place in the Early Middle Ages, and it would not do so until the 12th century (Evans 2001, p. 13). However, examining this in parallel with the etymology of the word "philosophy" will provide a better insight to understand the subject. Thus, it is possible to say that the late distinction between religion and philosophy that Evans (2001) mentioned resulted from the word "philosophy" being seen literally as the love of wisdom. Christians deemed their religion the source of knowledge and defined themselves as lovers of knowledge in the patristic era. According to them, Christianity was the real philosophy. For instance, Clement of Alexandria, the first patristic writer to introduce this notion, claims that the true philosophy was shown by the Son (Migne 1715, *Stromata*, I, 18: 90), and Augustine mentions Christianity as the real philosophy (Migne 1841b, *Contra Iulianum Pelagianum*, IV, 14: 72).

## 2. Fundamental Notions concerning Early Christian Philosophy

Upon Jesus's death, his apostles took on the mission of explaining and spreading Christianity. The period in which the apostles and their fundamental works appeared as a result of these efforts is known as the apostolic era (or period). The period between the end



of the 1st century and the 8th century, after the apostolic era, is referred to as the patristic era (or period). Patristics is also a field generally thought to apply to the lives of Church fathers, although it could equally apply to the less visible Church mothers, their works and doctrines within early Christianity or the history of the Church. The word "patrology", which was coined in *Patrologia*, published in 1653 by one of the most important Lutheran theologians of the 17th century, Johann Ernst, the son of Johann Gerhardt, after the death of his father, is used identically with patristics (Schmeling n.d., p. 8). However, today, patrology is used mostly to define edited collections of the works of the Latin and Greek Church fathers. Jacques-Paul Migne's collections entitled *Patrologia Latina* or *Patrologia Graeca* can be given as an example of these. In terms of literary history, early Christian or ancient Christian literature also describes the authors and their works.

As for philosophy, the early period of philosophical activities, which is defined as "Pre-Medieval Influences" by Copleston (1993) and as "Medieval Philosophy" by Gilson (1955), is known as "Patristic Philosophy" within the history of philosophy. The authors within the scope of patristics, patrology or patristic philosophy are frequently referred to as Church fathers or Church doctors. Although these titles are mostly considered identical, there are some differences between them. The title "Church father" was given to bishops who were the heads of the Church when the Church first emerged as an institution, and these bishops functioned as educational and doctrinal authorities. Later, the title of "Church father" was given to authors who assumed the roles of advocates and narrators of faith even though they were not members of the clergy. Hamell mentions that there were four conditions to obtain the title of "Church father": doctrinal integrity and soundness, the sanctity of life, approval by the Church and having lived during the early period of the Church (Hamell 1968, p. 9). We also learn from Hamell that some writers who partially fulfilled these conditions—such as Tertullian, Origen and Eusebius of Caesarea[3]—were given the title of "Church father" (Hamell 1968, p. 9). Another title, as important as "Church father", was "Church doctor", and this was given to the authors who met the higher conditions of being a "Church father". Eight of these authors, who had a deeper knowledge of the Christian doctrine and internalized a steady faith and the sanctity of life, are known as the "eight great Church doctors": Basil the Great, Gregory of Nazianzen, Chrysostome, Athanasius, Ambrose, Jerome, Augustine and Gregory the Great. There are female Church doctors and mothers, as well as male Church doctors. The title "Church doctor" was given only to four women by the Church: in the year 1970 to Theresa of Avila[4] and Catherine of Siena[5], in the year 1997 to Theresa of the Child Jesus[6] and in the year 2012 to Hildegard of Bingen[7]. On the other hand, the title "Church mother" is given by some modern scholars to women who made great contributions through the teaching and spreading of Christianity during its early period and who were even sometimes martyred for the sake of this ideal. Saint Macrina and Saint Monica, as Church mothers, are of great importance to patristic philosophy. While Macrina and Monica are among the philosophers whose views we can attain through the accounts written by men who were close to them, there is an ongoing debate about whether these accounts are factual. In fact, during the apostolic or patristic era, "women are mostly represented by men writing about them or quoting them" (Cohick and Brown Hughes 2017). The major reason behind this is the restrictive attitude concerning the social lives and education of women in interpreting the Christian doctrine. Nevertheless, among the Church mothers, Macrina and Monica, who are represented by the important writers of their era and could be described as philosophers, according to the very well-known criteria of patristic philosophers, are quite valuable to the Mediterranean Region.

## 3. Church Mothers and Their Way of Philosophizing

### 3.1. Saint Macrina

Saint Macrina (c.330–c.379) was the sister of the Cappadocian Church fathers, Saint Gregory of Nyssa and Saint Basil the Great (Basil of Caesarea). Both of her brothers mention Macrina's education and knowledge in their works. But Gregory wrote works[8] in

which he narrates Macrina's life as a philosophical biography (Silvas 2008) and deathbed philosophical discussion, similar to Plato's *Phaedo*. In his famous *Letter 19*, he glorifies Macrina with quotations from the Scripture as "a strong tower", "a shield of favor" and "a fortified city", and he also acknowledges her as a teacher and mother (Silvas 2008, p. 87). Silvas (2008) rightfully claims that this letter is a precursor to the *Life of Macrina* and *On the Soul and the Resurrection*. In the *Life of Macrina*, beginning with a narration of Macrina's life under both her known and secret names, Gregory first mentions their paternal grandmother, Macrina the Elder. He explains that Macrina's name was taken from her grandmother but that a few people knew that she also had another name. Her mother, "not being able to take a vow of virginity, was forced to marry someone because of her beauty" and, sometime later, became pregnant with Macrina. When the time had come for her mother to give birth, she saw a person with a marvelous appearance in front of her. This person told her that the child she was carrying in her arms had the name Thecla[9] before suddenly disappearing (Silvas 2008, pp. 111–2). With this apparition, Gregory divinely identifies Macrina as the most important woman martyr and transfers onto her all of Thecla's attributes. Although Thecla managed to survive the beasts' attack, in the fourth century, she became an object of devotion as a martyr in much of Asia Minor (Muehlberger 2012). This identification could be interpreted as the Christian version of the pagan katasterism or deification[10] (Sunberg 2017), through which Gregory highlights and celebrates all the necessary virtues for his ascetic community. That interpretation may be a positive contribution to Muehlberger's (2012) claim regarding Gregory's works as a Christian project of cultural reclamation. Apart from this, Gregory portrays Macrina as a Christian philosopher and, by doing so, defines Christian philosophy as a Christian way of life. He says, "[Macrina] had raised herself through philosophy to the highest summit of human virtue" (Silvas 2008, p. 110). He uses the word "philosophy" to denote the ascetic lifestyle of the Christians (Silvas 2008).

After establishing a correlation between Macrina and Thecla, Gregory stresses the philosophical inclinations of Macrina. Then, he resumes the biography by narrating her education, intelligence and the fact that she followed in the footsteps of Thecla with regard to marriage. Macrina's mother taught her not the "degenerative verses" of writers of tragedy or comedy based on secular primary education, as it was the tradition to do, but verses on morality from the Holy Scriptures (Silvas 2008, p. 113). However, we learn from Gregory that Macrina internalized these verses, rather than memorizing them. Her father wanted her to marry a well-educated and noble man, and at the age of twelve, he betrothed her to someone suitable, in his own opinion. Nevertheless, when Macrina's fiancé died before their marriage, her father and many people in her family made efforts to search for another proper suitor. Macrina did not allow them to convince her to marry anyone, as can be seen by her claim: "since by nature marriage is but once only, as there is one birth and one death. . .". She insisted that "he who had been joined to her by her parents' decision had not died, but that in her judgment he was alive to God (Luke 20.38, Romans 6.11) through the hope of the resurrection (Acts 23.6), and was away on a journey, rather than dead, and that it would not be right to keep faith with one's bridegroom who had gone on such a journey" (Silvas 2008, pp. 115–6).[11] At this point, it is worth remembering that Macrina was twelve years old. The connection she established with what she learned reveals the quality of her education and intelligence. Gregory states that Basil the Great had a condescending attitude towards the leading figures of his field with his vanity stemming from his talent after he returned home from his rhetorical education. At this point, too, Macrina stepped in and convinced Basil the Great to renounce all material things, leading him to philosophy (Silvas 2008, p. 117). Likewise, she led her youngest brother, Petrus, to philosophy by undertaking the responsibility for his care and education (Silvas 2008, p. 123). Petrus (Peter of Sebaste) was appointed bishop, like his brothers. Shortly after her father's death, Macrina "became the spiritual mentor of the household, educating, comforting, and developing a devoted community" (Cohick and Brown Hughes 2017).

Discussing a philosophical topic even on her deathbed, Macrina had a philosophical conversation with Gregory about the soul and explained the reason for a life led in a

body during her very last hours. She explained how humans were created and became mortal, when death would arrive and when the resurrection would take place (Silvas 2008, p. 129). It is important for Macrina's identity as a philosopher that Gregory described her philosophical speech on the soul as *περί τε τῆς ψυχῆς φιλοσοφοῦσα*, [12] meaning that what Macrina did was philosophize on the soul. That discussion was written by Gregory in the style of a Socratic dialogue. According to Muehlberger (2012), Gregory identifies Macrina with Socrates as a part of his project concerning a cultural reclamation. She claims that Gregory aims to validate Socrates for Christians through his sister, as opposed to Julian, linking someone's identity with their choice of books. Although this claim successfully explains the cultural and political conflicts of the era, it is not related to the reality of Macrina's identity. That is because, if the facts about Macrina were not necessary for this fiction, Gregory could have written about Macrina the Elder or his father. So, Macrina's character in Gregory's works may be a real woman without damaging the motive for Gregory's project. Moreover, Corrigan (2013) also claims that Gregory's works are important documents and that Macrina, as narrated in these works, reflects a real woman who is not completely fictional by the same logic.

Addressed by Gregory as *ή ἀδελφή καὶ διδάσκαλος* [13] (Silvas 2008, p. 171), Macrina defended the immortality of the soul in the work entitled "*On Soul and Resurrection*". In this work, Macrina, who had a deeper knowledge of the Epicurean and Stoic philosophies and explained these philosophers' views on the soul and the body, suggested that the soul was one and, for that reason, could not be divided; and since it did not disappear after death, it was immortal. Gregory started his work with the death of their brother, Basil the Great, and Macrina commenced her dialogue by asking what death actually was and why it caused grief. She expressed that Epicurus had reduced the nature of existing things to the phenomenal: "[Epicurus] He made our senses the only measure by which things are to be comprehended. He shut down completely the sensing capacities of the soul and was incapable of contemplating anything intellectual and bodiless. It is just like someone confined in a hut who remains unable to see the wonders of the heavens because he is cut off by the walls and roof from seeing what is outside" (Silvas 2008, p. 175). For Macrina, Epicurus and other pagan philosophers could not comprehend the divine operation, divine power or order. Gregory asked Macrina what could be learned about the soul from the things manifesting in the body, and Macrina stated to him that humanity was a micro-cosmos and the universe was a macro-cosmos. Macrina argued that the soul was nonmaterial, bodiless and active and that it gave away the evidence of its movements through its organs, acting by its nature. For her, the fact that there was no movement in the body after death indicated that the soul had already left the body. The soul only moved when there was sense in the organs, while mental power moved by sensing this. Progressing to the topic of the formation of movement after discussing the process of the senses, the compatibility between the things perceived by the senses and reality, Macrina attempted to express what the soul was not using negative theology: "But we learn much about many things in just such a way, for we interpret the actual being of whatever it is we seek by affirming what it is not" (Silvas 2008, p. 183). When the topic evolved into the connection of the emotions and passions to the soul, Macrina argued that these were the effects of nature (*πάθη*/pathē), not the essence of nature, since essence is something that is: "All these emotions are around the soul and yet they are not the soul, but only like warts growing from the soul's thinking part. They are reckoned to be parts of it because they are growing on it, and yet they are not what the soul is in its substance" (Silvas 2008, p. 192). Macrina, after defining emotions—especially anger and passion—separately in the essence of the soul, exemplified through the blacksmith and iron that emotion was not good or bad on its own and that that which made it good or bad was our intention, while accepting that God, the creator of everything and the Good, created good and bad emotions. Humans can shape their emotions in the way they like, just as the blacksmith gives form to iron in the way he wants. When Gregory brought the topic back to the soul and brought up another topic about where the soul would go after death, Macrina put forward the three

rational forms of nature: "one from the very beginning was allotted a bodiless life, and this we call the angelic; another is interwoven with the flesh, and this we say is human; a third is released by death from the things of the flesh, which is contemplated in the case of souls" (Silvas 2008, p. 199). For her, the souls that were supposed to go to the underworld, namely the third rational form of nature, were not obliged to go anywhere. Macrina equated the underworld with the soul leaving the body. She strongly opposed the idea of reincarnation, expressed that the resurrection had a condition of being born to the world and pointed out that that was the soul's way of coming back to its primitive state. She went on to exemplify the soul's return to its primitive state with the seed sown into the ground. According to her, Adam was the first spike, and human nature was divided into a multitude by original sin. When humans would be resurrected, they would be a myriad of spikes in the corn fields instead of a single spike.

### 3.2. Saint Monica

Saint Monica or Monnica (4th c.), the mother of St. Augustine, bishop of Hippo, one of the leading figures in the history of the Church and Christianity, was most probably born in Thagaste, like her son, and raised as a Christian. However, she married Patricius, a pagan official. Patricius converted to Christianity when Augustine was almost fifteen years old. Monica and Patricius had three children, Navigius, Perpetua and the eldest, Augustine. We learn about both her life and her philosophical views from Augustine. After her husband's death, Monica went on a journey to Italy with her son Augustine, whom she wanted to convert to Christianity, and died on her way back to Africa in Ostia. Feeling the grief of her death deep in his heart, Augustine writes[14] about her value in his own life: "I will not refrain from saying whatever my soul feels about your maid who gave birth to me, both in the flesh to this temporal life and in the soul to the eternal Light" (Migne 1841a, *Confessiones* IX.8.17). When Monica, who was the child of a Christian family, reached the age of marriage, she married Patricius and served him "as her lord" (*Confessiones* IX.9.19). In an effort to convince him to convert to Christianity, Monica forgave his infidelities and never fought with him just so that he would believe in one true God. In any case, Patricius had a very short temper, and knowing this, Monica explained her views when he calmed down, and she judged many women who did not act in this way harshly. Still, she educated many women on how to treat their husbands through the example of her marriage. Although her mother-in-law initially treated Monica badly "because of the bad maids", she began to grow fond of her because of her actions, and they lived in peace with a mutual understanding of each other (*Confessiones* IX.8.20). Augustine had a discussion with his mother on her deathbed, just as Gregory had with Macrina. They talked about what kind of eternal life the saints would have, the source of life, the universe, how God created the universe perfectly, Israel and the Holy Spirit. Unfortunately, Monica died when her body could not find the strength to fight a fever that lasted for almost five days. However, before that, Augustine retreated to his friend Verecundus's house in Cassiciacum, accompanied by his mother (386), and here he stayed with his mother, Monica, his son Adeodatus, his brother Navigius, his cousins Lartidianus and Rusticus, and his two students Licentius and Trygetius. Meanwhile, with the help of his friend Alypius, he edited his works entitled *Against Academicians* (Contra Academicos), *Soliloquies* (Soliloquia), *On the Order* (De Ordine) and *On the Happy Life* (De Beata Vita), known as *The Cassiciacum Dialogues* by modern scholars (Foley 2019, pp. xxiii–xxiv). In his *On the Order*, when his mother asked how they were and how far they had progressed in the middle of the discussion with Trygetius and Licentius, Augustine told her that she should come and join the discussion and her questions or answers would be written down, just like theirs. This statement shows both how much he valued his mother and how he supported the participation of women in philosophy to the point of being recorded. At that time, Monica refused, saying that she had not encountered women in such a discussion in any of her son's books. However, Augustine stated that those who opposed the participation of women in philosophical discussions were arrogant and ignorant, that they paid attention

not to human qualities but to clothes and vanity and that they read without understanding why what they were reading had been written (I.11.31). Pointing out the very existence of many women philosophers, Augustine mentioned that he loved her philosophy the most: "...et philosophia tua mihi plurimum placet" (I.11.31). Afterwards, he explained to his mother what philosophy was: "hoc graecum verbum quo philosophia nominatur, latine amor sapientiae dicitur (I.11.32)".[15] Then, remarking that Jesus Christ had made a distinction between the material-world philosophers and the otherworldly philosophers and that the material world was not important, he stated that his mother loved philosophy more than him. However, Monica refused, saying Augustine had "never lied like this before" in a "motherly" manner. After that, the delayed discussion took place, and at some point, Monica was included, together with her opinions. Licentius claimed that there was an order in God's deeds and that there was nothing that was not God's act; hence, there could not be anything outside the divine order. Augustine opposed by saying that evil deeds might come from God: "In that case, if God is always just, he will always be good and evil" (II.7.22). Thereupon, Monica intervened and suggested that there was not any judgment of God in the absence of evil and that God could not be defined as just as long as he did justice to both good and evil. This time, Augustine claimed that evil would take its form outside the order, for they would have accepted the idea of evil created by God if the views of Licentius and Monica were to be accepted. At the point where Licentius seemed not to understand and to remain silent, Monica argued that she opposed this view, arguing that there could be nothing outside of God's order, that evil did not remain outside of the order of God's justice and that, by reducing evil, God made it a part of the order (II.7.23–24).

In another work of Augustine's entitled *On Happy Life* (De Beata Vita), we see that Monica took an active part in the ongoing discussion. Monica, agreeing with Augustine that the body was nourished by food and the soul by knowledge, argued that the mind was nourished by its theories and thoughts in answer to Trygetius, who disagreed with her. On the matter of happiness, on the other hand, Augustine responded to Monica's answer that if a person wanted good things and had them, they would be happy, by saying that she had completely conquered the castle of philosophy (arcem philosophiae tenuisti). Quoting from *Hortensius* by Cicero, Augustine said the following on Monica's reaction, after stating that her thought was wrong: "My mother howled at these words so much that we thought a great man was sitting among us, completely forgetting her gender" (II.10.9–11). Monica, who established a link between want and need and happiness and unhappiness, argued that a person who was always in need was miserable. She then claimed that those who knew God and led good lives had befriended God by making a connection between happiness and God but that those still searching for God had either befriended Him or become His enemies. While the initial arguments of Monica were approved by Augustine and the others, he found the questioning by Monica, when she was left unsatisfied, admirable and was delighted that she had expressed the exalted thing he had planned to put forward at the end and which had been mentioned by the philosophers in their works. Monica argued that a wealthy person felt the need for wisdom in the event that they feared losing what they had at a time when they did not have any desire for anything else. The group then began to discuss the matter of wisdom, and when Augustine stated that a happy life was a pious and perfect knowledge of God, who guided people to the Truth, Monica responded that this kind of life could be ensured by having sound faith, living hope and goodness.

## 4. Conclusions

When it becomes necessary to position these two Mediterranean Church mothers in the history of patristic philosophy, they deserve to be acknowledged as philosophers by the Christian philosophy of their eras. As explained by Augustine to his mother, philosophy essentially is *philosophia* in Greek and *amor sapientiae* in Latin, meaning the love of wisdom. This wisdom belongs to the Holy Scripture—God, as far as Christianity is

concerned; the Christian who possesses this wisdom has true knowledge and is the one who is the real philosopher, having the right answer, in contrast to the other philosophical systems. Hence, patristic philosophers made efforts to interpret the knowledge provided by the Holy Scripture rationally and yet defined the Holy Scripture as a set of methods to attain happiness. As also mentioned by Sadler, philosophy is a way to essentially lead an intellectual life in both the past and the present (Sadler 2007, p. 10). Macrina and Monica led exemplary lives for all Christian women by having the Holy Scripture as their guide and elevating the level of their education at every single opportunity. Although they did not establish any philosophical system, they became representatives of Christian philosophy and, hence, of patristic philosophy via their approach to philosophy, which was suitable for their era. Furthermore, the fact that they were not the authors of any work, that the men writing about them idealized them to various degrees and that these male authors used them as teaching materials do not change the fact that they were Christian philosophers. One should also keep in mind the example of Socrates and the restricted lives of women. Moreover, every character is a mix of fiction and reality, and every author has a motive that could almost be considered propaganda. That is to say that, although claims like Muehlberger's (2012) imply that under this idealization lies a purpose which imperils the historical reality of these characters, it is evident that a project or piece of propaganda could not have worked in that time without facts. Apart from their reality in works, their participation in philosophical discussions show their commitment to the Scripture, but at the same time, it also shows that they possessed the element of philosophical speculation and inquiry. Gregory's and Augustine's works are fictions on some level since every written record, document or literary work could be considered fiction—the process of revealing an idea from the mind to something permanent and accessible to others may be compared to the meaning of the word $\lambda\acute{o}\gamma o\varsigma$/logos. In Liddell's and Scott's lexicon (Liddell and Scott 1983), it has two synonyms in Latin: first, *vox* and *oratio*, and second, *ratio*. This grouping demonstrates the difference of logos from $\mu\bar{\upsilon}\theta o\varsigma$/mythos. Mythos means "anything that is delivered by the word of mouth", whereas logos involves a filtering process. Mythos could mean gossip or a tale, whereas logos always includes logic or reasoning. Once an idea or, basically, a sentence is written down, one should devise the setting by filtering down their thoughts. If reality is in question, we believe no written word completely reflects the actual version of the author's mind since every author has a persona and motive to write. Therefore, Macrina's and Monica's alleged words could be works of art with fictitious elements on some level, but they indicate their reasoning and way of discussing. They demonstrate that Macrina and Monica inquire and carry out their lines of reasoning by always adhering their ideas to the Scripture, just like the other patristic philosophers.

**Author Contributions:** Conceptualization, E.K.I. and R.A.; Data curation, E.A., N.K.P. and A.B.Ö.; Formal Analysis, E.K.I., R.A. and N.K.P.; Investigation, E.K.I. and R.A.; Methodology, E.K.I., R.A. and A.B.Ö.; Visualization, E.A.; Writing—original draft, E.K.I., R.A. and A.B.Ö.; Writing—review & editing, E.K.I., R.A., E.A. and N.K.P. All authors have read and agreed to the published version of the manuscript.

**Funding:** This research received no external funding.

**Institutional Review Board Statement:** Not applicable.

**Informed Consent Statement:** Not applicable.

**Data Availability Statement:** The data presented in this study are available within this article.

**Conflicts of Interest:** The authors declare no conflict of interest.

## Notes

1    It was forbidden to convert to Christianity during the reign of Nero: "Christianos esse non licet" (Schaff 1998).
2    The word "ἐκκλησία" (ekklēsia) in Ancient Greek essentially means assembly. Hence, places called house churches were assemblies scarcely institutionalized and yet organized according to its own rule.

3　Even though Hamell mentions Tertullian, Origen and Eusebius of Cesaria as Church fathers, they are not considered Church fathers officially by the Church.

4　27 September 1970. Proclamazione di Santa Teresa D'Avila Dottore della Chiesa: Omelia del Santo Padre Paolo VI. Vatican: The Holy See. Available online: https://www.vatican.va/content/paul-vi/it/homilies/1970/documents/hf_p-vi_hom_19700927 .html (accessed on 30 January 2022).

5　3 October 1970. Proclamazione di Santa Caterina Da Siena Dottore della Chiesa: Omelia del Santo Padre Paolo VI. Vatican: The Holy See. Available online: https://www.vatican.va/content/paul-vi/it/homilies/1970/documents/hf_p-vi_hom_19701003 .html (accessed on 30 January 2022).

6　20 October 1997. Address of the Holy Father Pope John Paul II to pilgrims in Rome for the proclamation of Saint Thérèse of the Child Jesus and the Holy Face as a doctor of Universal Church. Vatican: The Holy See. Available online: https://www. vatican.va/content/john-paul-ii/en/speeches/1997/october/documents/hf_jp-ii_spe_19971020_teresa-lisieux.html (accessed on 30 January 2022).

7　7 October 2012. Apostolic letter: Proclaiming Saint Hildegard of Bingen, professed nun of the order of Saint Benedict, a Doctor of the Universal Church: Benedictus PP. XVI. Vatican: The Holy See. Available online: https://www.vatican.va/content/benedict-xvi/en/apost_letters/documents/hf_ben-xvi_apl_20121007_ildegarda-bingen.html (accessed on 30 January 2022).

8　For Gregory's works, Silvas's translation is followed.

9　Saint Thecla tried to spread and teach Christianity with Paul during the apostolic era of Christianity. Thecla, as the child of a noble and rich family, being engaged to a noble and rich man, was, so to speak, taken with the doctrines from the moment she heard Paul talk about virginity, insomuch that she gave up eating and drinking for three days straight and never talked to either her mother or her fiancé. She could not resist listening to Paul; even when her fiancé put him in jail, she went directly to his cell. She attempted to be burned because of her mother's pleas to "burn my daughter as a lesson to the other women", but she was saved by a miraculous rain. After leaving with Paul for Antioch, Thecla was sentenced to death by being thrown to wild animals in the arena for the defamation of another man. Although she encountered a lot of wild animals, she survived miraculously and made a lot of people, especially women, convert to Christianity. Her account was collected in a work titled *The Acts of Paul and Thecla*, which was proclaimed apocryphal afterward. Tertullian is the first person to claim that this work was apocryphal, and he stressed that it was not appropriate for women to teach. For detailed information about Thecla, see Aquilina and Bailey (2012). Tertullian believed that women should not be teachers, cf., 1 Tim. 2: 12.

10　For the Christianization of deification or theosis, see Sunberg (2017).

11　While the earliest legal age for marriage was 12, girls were usually married at the age of 16. It could be assumed that the thing that made this system continue was suspicion towards girls' fathers or guardians, as if they were in a sort of political or financial conspiracy (Clark 1994, p. 14). That could be why Macrina's father and her relatives insisted on marrying her. Clark attributes Macrina's claim that the betrothal was identical to a marriage, as the betrothal was deemed a serious commitment (Clark 1994, p. 14).

12　"Peri te tēs psykhēs philosophousa". Gregory of Nyssa (n.d.). *The life of St. Macrina-Excerpts*. Early Church Texts. Available online: https://earlychurchtexts.com/main/gregoryofnyss/life_of_st_macrina.shtml (accessed on 8 February 2022).

13　Hē adelphē kai didaskalos (sister and teacher).

14　Augustine (1992), Augustine (2007) and Knöll (1962) are followed.

15　"the word philosophy is Greek and means love of wisdom in Latin".

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
