# Peer review of "Two Mediterranean Church Mothers: Their Presence and Importance in Patristic Philosophy"

_religions, doi:10.3390/rel14101220_

Round 1

Reviewer 1 Report (Previous Reviewer 1)

Some amends to make:

l. 15 with> through the lens of

l. 17 basin of > crucible for

l. 63 disintegrating > disaggregating

l 81 As known, philosophia consists of a combination formed with φιλεá¿–ν (philein) meaning “to love” and σοφία (sophia) meaning “wisdom” in Ancient Greek and is translated as “love of wisdom”. repeats l. 1. transfer to opening of article.

l. 96 make more of  and Church mothers  perhaps with an added note, as indicated here:

Patristics, also, is a field generally thought to apply to the lives of Church fathers, although it could equally apply to the less visible Church mothers,

l. 127 This saint is always Theresa of the Child Jesus

l. 156  human> person/vision/apparition

l. 158 with the most important woman martyr and transfers onto her all Thecla's attributes

l. 187 for his own opinion > according to

l. 281 she married him

l. 282. rewrite: In efforts to convince him to be a Christian, Monica forgave his infidelities and never fought with him just for him to believe in real God. delete real, add one, true God. The highlighted section does not make sense.

l. 321 delete herein- never used. 

l. 357 the Christian who possesses this wisdom, has true knowledge and is the one who is the real philosopher because he or she has the right answer in contrast to philosophers who belong to other philosophical systems. amend.

check all references as these are not in the correct format

l. 471 Belgium: Brepols> Turrhout: Brepols

similarly Minnesota, Oregon, Michigan all states rather than cities.

ie Cohick and Brown Hughes should be Ada, MI: Baker Academic

Further proofreading and pre-publication checking of the English is essential for this article as there are numerous details which cause problems for the reader.

Author Response

We thank the reviewer for their time and effort. The manuscript is professionally proofread and edited. All notes were taken into consideration and the necessary adjustments and corrections were made. 

Reviewer 2 Report (Previous Reviewer 2)

Thank you for the opportunity to read this paper. It is certainly an improved version of the previous attempt, however I still think that it needs some more work, at least in terms of expression. 

I have tried to be as helpful as possible, so below are the corrections that I think could be made to improve it.

Given than it is the second time that I am reading this article, I think that if the publisher decides to send it back for more corrections, it would be better for a fresh pair of eyes to look at it at the next go.

My general comment is that it is an interesting topic, however it could have been executed with greater clarity and precision. 

With regard to Section 3, I think that Macrina and Monica should be clearly distinguished by having separate sections/subsections devoted to them. Also, the material on Macrina is much longer than the material on Monica. This may be inevitable and may not be a problem, however this should be recognised and explained by offering the reason for this asymmetry (lack of sources on Monica, or whatever other reasons there may be). 

This is the list of particular corrections that I would like to suggest is (I go by lines numbers):

7-9: I would rephrase, reads a bit clumsy

29-30: rephrase

38: ‘synonymously’ is redundant, I think

44-46: rephrase

46: who is ‘they’? rephrase

50: change ‘conclusions’ to ‘results’?

53: who is ‘he’?

64-69: long sentence, rephrase

71-72: themselves?

73: themselves?

78-79: rephrase

79: Evans is given a page number, but other sources are usually not

83-90: I am not sure how this follows, or explain the ‘late distinction’

94-96: references?

106: Gilson describe ‘early period of philosophical activities’ as ‘Medieval Philosophy’…? Somehow I doubt this (it makes little sense), by who knows?

126-128: why not list these chronologically?

132-134: rephrase

144-146: rephrase

158-159: rephrase (unclear)

181: as was the tradition

184-185: suitable for his own opinion?

187: rephrase

195: is ‘Basil’ Basil the Great?

200-201: any reference to this canonisation?

205: change ‘talk’ to ‘conversation’

206: humans were created

209: translate the Greek?

210-220: not a very strong section, and not an easy read

224-225: rephrase

228: ‘phenomenon’ rather than ‘phenomenal’?

251-254: rephrase

263: the idea of reincarnation

268: remove ‘the’ from before ‘humans’

270-72: reference?

283-284: rephrase

317-318: unclear

350: lived hope? (not ‘live’)

358: ‘possess this wisdom’ is better than ‘has’

363: remove comma from before ‘and’

367: remove ‘the’ at the end of the line

368-371: rephrase

379-389: not a great section

426: ‘proclaimed apocryphal afterward’?

Improved, but I think in need of further improvement if it is to be published in a peer-reviewed journal

Author Response

We thank the reviewer for their time and effort. The manuscript is professionally proofread and edited. All notes were taken into consideration and the necessary adjustments and corrections were made. 

This manuscript is a resubmission of an earlier submission. The following is a list of the peer review reports and author responses from that submission.

Round 1

Reviewer 1 Report

This article has at its heart an interesting narrative which needs to be told about the women who contributed to philosophical thinking in the early period. Unfortunately it is hard to follow because the English style is poor. The authors need to have the article thoroughly revised by a native speaker. There are basic grammar errors throughout:

cf l.110 were > was

stylistic errors

cf. l.143 got pregnant > became pregnant

The authors should consider giving a fresh title: Wise Women acknowledged in Patristic Philosophy (?) as the present one is unclear.

The abstract needs revising. The opening sentence does not make sense and does not set the scene. As a whole it does not sell what is discussed in the article. Put more emphasis on the methods and texts you plan to use.

The premise that little research has been carried out could be revised by taking account of the following studies which are not mentioned:

Sunberg, Carla D. The Cappadocian Mothers. 1st ed. Cambridge: James Clarke &, 2018. Web. It would be good to also give an overview of Aquilina & Bailey and set out what it covers. It is cited but only as a passing reference.

There are a number of relevant works on the early Church mothers which need acknowledging, citing:

Consider other writings about the women, though not about their philiosophy, still relevant ie

On Monica: Sehorn, John. "Monica as Synecdoche for the Pilgrim Church in the Confessiones." Augustinian Studies 46.2 (2015): 225-48. Web.

On Macrina: Muehlberger, Ellen. "Salvage: Macrina and the Christian Project of Cultural Reclamation." Church History 81.2 (2012): 273-97. Muehlberger compares her with Socrates.

Frank, Georgia. "Macrina's Scar: Homeric Allusion and Heroic Identity in Gregory of Nyssa's Life of Macrina." Journal of Early Christian Studies 8.4 (2000): 511-30.

Corrigan, Kevin. "Syncletica and Macrina: Two Early Lives of Women Saints." Vox Benedictina 6.3 (1989): 241.

Dewhurst, E. Brown. "On the Soul and the Cyberpunk Future: St Macrina, St Gregory of Nyssa and Contemporary Mind/Body Dualism." Studies in Christian Ethics 33.4 (2020): 443-62. Web.

Emily Chesley. "The Mercy of Macrina the Younger." Studia Patristica. Vol. CXV - Papers Presented at the Eighteenth International Conference on Patristic Studies Held in Oxford 2019. Vol. 115. Peeters, 2021. 1. Web.

More worrying is the omission of the following major book which does examine Macrina as a philosopher: Silvas, Anna M., Macrina the Younger: Philosopher of God (Turnhout: Brepols, 2008).

It may also be worth considering this source both the original and the article about it: Krausmüller, Dirk. "The Encomium of Catherine of Alexandria ( BHG 32b) by the Protasecretis Anastasius, a Work of Anastasius “the Stammerer”." Analecta Bollandiana 127.2 (2009): 309-12. Web.

I think that using encyclopaedias as a source of information on the women is not the best approach.

Reviewer 2 Report

A very interesting article. However, the expression needs to be fixed in the process of editing. Here are just some examples of statements that are need of revision for greater clarity and better flow. There are many more, but I just list these for the purpose of illustration:

vv 20-21: Traditionally, we tend to answer this question with an etymological explanation suggesting that philosophy is the love of wisdom as we will below.

vv 36-38: Because philosophy promised a kind of happiness that could only be obtained in the material world, however Christians didn’t live for the material world that was temporary. 

vv. 46-50: Enumerating the differences between Christians and Greek philosophers as disciples of heaven versus disciples of Greeks, the ones who have business with life versus the ones who have business with fame, the ones who deals with facts versus the ones who deals with words, and so on (XLVI. 18), he positions the philosophers as thieves, and people distorting the truth and using it for their own benefit.

vv. 75-77:  Christians deemed their religion as the source of knowledge and defined themselves as lovers of knowledge, hence wisdom for they followed this religion in patristic era.

vv. 86-88: Patristics, also, is a field concerning the lives of Church fathers and Church mothers, their works and doctrines within early Christianity or Church history and deemed as a branch of theology.

vv. 103-105: Afterwards this title was given to the authors assumed the roles of advocates and narrators of faith even though they were not a member of the clergy.

etc. etc.

With regard to the content, I have just a few notes:

- in v. 78ff the author writes about St Augustine's appreciation of philosophy and its place in Christianity: "... clearly suggested that Christianity was the real philosophy..." (vv. 79-80). However, it was Clement of Alexandria who first introduced this notion some 200 years before Augustine, see his Stromata I, 18, 90.

- in v. 108ff the author writes that Tertullian, Origen and Eusebius of Cesarea are considered by Hamell as 'Church fathers" - this may be true, but ultimately the Church does not consider any of them as a Church father, so I think that this should be stated, at least in a footnote. 

- v. 134 - "Saint Macrina, who is an Anatolian as well..." - what does this mean? What/who is an Anatolian?

v. 149 - I think that the material on St Thecla should be preceded, not followed, by the statement that these accounts about her are apocryphal. They sound very mythological, I think the author could warn the reader in advance that they may not be historically accurate or true.

Reviewer 3 Report

Right from the start, the title is poorly written ("the" is not appropriate) and the first 3 lines of the abstract (lines 5-7) do not make any sense.  This is very poorly written (with far too many grammatical mistakes for me to list), to the point that I do not know what the paper is trying to demonstrate.